# Details of the Contents of Paranoid Thoughts in Help-Seeking Adolescents with Psychotic-Like Experiences and Continuity with Bullying and Victimization: A Pilot Study

**DOI:** 10.3390/bs10080122

**Published:** 2020-07-29

**Authors:** Gennaro Catone, Antonella Gritti, Katia Russo, Pia Santangelo, Raffaella Iuliano, Carmela Bravaccio, Simone Pisano

**Affiliations:** 1Department of Educational, Psychological and Communication Sciences, Suor Orsola Benincasa University, 80132 Naples, Italy; antonella.gritti@unisob.na.it (A.G.); katiarusso83@hotmail.it (K.R.); 2Department of Neuroscience and Rehabilitation, AORN Santobono-Pausilipon, 80129 Naples, Italy; piasantangelo82@gmail.com (P.S.); pisano.simone@gmail.com (S.P.); 3“Ospedale del Mare” Hospital, 80147 Naples, Italy; raffaella.iuliano2@virgilio.it; 4Department of Translational Medical Sciences, Section of Pediatrics, Federico II University, 80131 Naples, Italy; carmela.bravaccio@unina.it

**Keywords:** psychosis, paranoia, adolescents, bullying, victimization, psychotic-like experiences

## Abstract

**Background**: Psychosis recognizes an interaction between biological and social environmental factors. Adversities are now recognized to be consistently associated with psychotic-like experiences (PLEs). The purpose of this study was to describe the contents of paranoid symptoms and to focus on their relationship with bullying and victimization in help-seeking adolescents. **Methods**: Help-seeking adolescents who screened positive for PLEs participated in the study. They performed a battery self-report questionnaire for data collection (paranoia: the Specific Psychotic Experiences Questionnaire (SPEQ); the content of paranoid thoughts: the Details of Threat (DoT); bullying victimization: the Multidimensional Peer Victimization Scale (MPVS); depression: the Children’s Depression Inventory (CDI); and anxiety: the Multidimensional Anxiety Scale (MASC)). **Results**: The participants were 50 adolescents (52% female; mean age: 170 months). The contents of their paranoid symptoms were related to victimization and, in particular, the certainty of threats was correlated with physical (0.394, p < 0.01) and verbal bullying (0.394, p < 0.01), respectively. The powerfulness of the threats correlated with verbal victimization (0.295, p < 0.05). The imminence of the threats was linked to verbal (0.399, p < 0.01) victimization. Hours under threat correlated with verbal (0.415, p < 0.01) victimization. The sureness of the threat had a moderate correlation with physical (0.359, p < 0.05) and verbal (0.443, p < 0.01) victimization, respectively. The awfulness of the threat was linked to social manipulation (0.325, p < 0.05). **Conclusions**: We described the content of the persecutory symptoms. The powerfulness, imminence, sureness, and awfulness of threats correlated with the level of physical, verbal and social manipulation victimization. Teachers and family must actively monitor early signs of bullying victimization, and school psychologists should promote preventive and therapeutic intervention. From a social psychiatry perspective, the prevention of bullying victimization is necessary.

## 1. Introduction

Psychosis aetiology recognizes an interaction between the biological and social environmental factors that together contribute to the onset and predict the course of the disorder [1]. While biological factors are difficult to manage, environmental ones are more easily recognizable and, moreover, they can be a target of prevention and treatment [2]. Among them, the occurrence of childhood adversities has recently been recognized as one of the environmental factors most associated with psychotic disorders [3,4], as well as with psychotic-like experiences (PLEs), a rather common dimensional phenomenon in adolescent people [5,6]. Among childhood adversities, one of the most extensively studied and more consistently found factors associated with impacting the well-being of children and adolescents is bullying and victimization, which consists of intentional and repeated aggressive acts toward the victim in the context of an imbalance of power [7,8,9]. Several studies have demonstrated that bullying and victimization represent a clear risk factor for psychotic symptoms; in the 2000 and 2007 Adult Psychiatric Morbid Survey, bullying and victimization were associated with paranoid ideation and hallucinations in cross-sectional and longitudinal analyses [7]. There is also evidence that specific early-life adversities, such as bullying and victimization, lead to specific psychotic symptoms, such as paranoia or hallucinations [10]. The prevention of bullying and victimization in schools is a sensible topic for the promotion of improved mental health in adolescents. Anti-bullying intervention effectively reduces school bullying [11]. 

There is evidence on the thematic continuity (thematic link) between the stressful experiences of life and the contents of delusional thinking [12]; furthermore, stressful experiences could not only play a role in the development of psychotic symptoms, but also may shape their contents and form [12,13]. From this perspective, in a developmental social psychiatry framework, verbal and social bullying and paranoia are strictly associated, as seen in a sample of help-seeking adolescents who have had PLEs [14]. To continue on this path, we used a model that includes not only the presence of paranoid thoughts but also the content of those thoughts. Freeman et al., in order to assess the details of delusional thoughts (e.g., conviction, distress, and the power and imminence of the threat), designed a new questionnaire: “The Detail of Threat”. In their study, the authors found that several features of the contents of the delusional thoughts were associated with depression, self-esteem, and anxiety. This questionnaire is complementary to a more quantitative questionnaire examining the presence of paranoid symptoms, such as the paranoia scale of the Specific Psychotic Experiences Questionnaire (SPEQ) [15]. There are several questionnaires that assess bullying victimization [16]. The Multidimensional Peer Victimization Scale (MPVS) [17] is a simple, fast, and reliable analysis tool that has been used in several studies previously [18,19].

In this framework that mixes a social perspective and developmental psychopathology, we conducted the present study that focuses on the details of the contents of paranoid thoughts. We explored the relationship of the latter with bullying victimization, also accounting for emotional symptoms. In fact, according to the cognitive model of psychotic symptoms, anxiety and depression play a decisive role in their onset and maintenance [20].

More specifically, the first aim of this study was to describe the details of paranoid symptoms in a cohort of help-seeking adolescents with PLEs. The second aim focused on revealing the relationship between some of the described details of the threat (i.e., the imminence, power, and awfulness) and bullying and victimization, controlling for anxiety and depression.

## 2. Materials and Methods

### 2.1. Study Design and Sample

This study had a cross-sectional design. The setting was the child and adolescent unit of the Department of Mental Health, Physical and Preventive Medicine of The University of Campania “Luigi Vanvitelli” in Naples. Help-seeking adolescents, screened for the presence of PLEs, constituted the original sample. The inclusion criteria were positive screening for PLEs and age between 12 and 18 years. The exclusion criteria were diagnosis of intellectual disability, autism spectrum disorder, a known diagnosis of schizophrenia, the presence of genetic or neurological disease and ongoing psychopharmacological treatment. After the screening, we had 50 participants (52% female, mean age 170 months ±18.4) who performed a battery self-report questionnaire for data collection. Patients and their parents, previously informed of the aims and methods of the study, signed an informed consent and the study was approved by the ethical committee of the University of Campania ‘Luigi Vanvitelli’ (N.499/29 April 2016). 

### 2.2. Measures

Screening instrument: The Adolescent Psychotic-Like Symptom Screener (APSS) was used to screen the initial population for the presence of psychotic-like symptoms. This questionnaire was composed of seven items with multiple-choice answers (0: no; 0.5: maybe; 1: yes). The cut-off score for the risk of PLEs was >2 points. The APSS had good predictive power, sensitivity, and specificity. A score of 2 or more had sensitivity of 70% and specificity of 82.6%. The item with the better predictive power was the question ‘‘Have you ever heard voices or sounds that no one else can hear?’’ with a positive predictive power of 71.4%, and a non-positive predictive power of 90.4% [21].

Content of paranoid thoughts: We used the Details of Threat (DoT) questionnaire, an instrument that assesses several details of persecutory symptoms, such as: imminence (when, where, and rescue factors), knowledge, power, sureness, awfulness, deservedness, unfairness, coping, and control regarding the threat. The questionnaire has been used by Freeman et al. in another study [22]. All the questions of the instrument are presented in the results. The DoT has questions with categorical (Do you know the person who is threatening = no, maybe, yes; When do you think the harm is most likely to happen? Please circle one of the time periods below = It has been happening recently/0 to 7 days/1 week to 1 month/1 month to 6 months/6 months or longer); ordinal (awfulness, coping strategies, deservedness of threat, unfairness, rescue factors, and control = possible answers from 0 to 10); and numerical (hours = possible answers 0–24; sureness = possible answers from 0 to 100). There is also an open-ended question (type of harm).

Dimensional paranoia: The dimensional level of paranoia was assessed using the paranoia subscale of the Specific Psychotic Experiences Questionnaire (SPEQ), derived from an adaptation of the Paranoia Checklist (PC), in which the last three items were omitted, and seven items were re-worded for the specific needs of this age range. SPEQ also included five other dimensions: hallucinations, cognitive disorganization, grandiosity, and anhedonia. In adolescents, the scale showed very good psychometric properties: Cronbach’s *α* of 0.93 [15] and Cronbach’s *α* of 0.85 [23].

Depression: We used the Children’s Depression Inventory (CDI) to detect depressive symptoms in the sample. The CDI is a 27-item questionnaire that measures the intensity of the depressive dimension. A score > 19 was a cut off for depression. The CDI is one of the most used tools in the field of child and adolescence psychiatry and it showed very good psychometric properties (Cronbach’s α = 0.80) [24].

Anxiety: We used the Multidimensional Anxiety Scale (MASC) to assess anxiety in the sample. MASC is a 39-item scale with five subscales: physical symptoms, social anxiety, harm avoidance, and separation anxiety. The score was obtained from different answers: Never true for me = 0, rarely true for me = 1, sometimes true for me = 2, and often true for me = 3. A higher score reflects a higher level of anxiety. MASC showed good psychometric properties (internal variability for the MASC was in the excellent range = 0.6–0.9) [25].

Bullying victimization: Bullying victimization was evaluated using the Multidimensional Peer Victimization Scale (MPVS). The MPVS is a 16-item scale with four subscales: physical victimization, verbal victimization, social manipulation, and attack on property. A score was achieved from three possible answers: Not at all = 0, once = 1, and more than once = 2. The total scale has a possible range of 0 to 32; whereas, the four subscales have a possible range from 0 to 8. Higher scores reflect more victimization. The MPVS showed a good reliability and validity to measuring bullying victimization (Internal consistency: Physical victimization = 0.85; Verbal victimization = 0.75; Social manipulation = 0.77; and Property attacks = 0.73) [17].

Other variables: The clinical records of each patient were inspected and several variables were extracted for descriptive purposes.

Statistical analysis: Statistical analysis was performed using the Statistical Social Packaging Software (SPSS) 20th edition. Descriptive statistics (means, standard deviations, ranges, and frequency distributions) were used in order to describe the clinical and socio-demographic characteristics of the sample, as well as the details of the threat variables. We performed a Spearman correlation analysis between all DoT items and the SPEQ paranoia scale. The Spearman partial correlation analysis was made between the contents of paranoid thoughts and bullying victimization (total and subscales), with anxiety and depression as covariates. The Detail of Threat variables were used both in the original form (partial correlation analyses with bullying victimization) and in the recoded form (in the descriptive analyses); in the latter case, all the ordinal variables 0–10 were transformed into categorical ones = no (0–3), perhaps (4–6), and yes (7–10).

## 3. Results

### 3.1. Descriptives

The participants included 50 adolescents (26 female, 52%). The mean age was 170 (±18.4) months (14 years and 1 month). The parent reported historic information included: 12% (n = 6) had birth problems, 18% (n = 9) and 24% (n = 12) had, respectively, psychomotor and language delay; 34% (n = 17) presented early separation anxiety; 34% (n = 17) presented a very poor school level, whereas 48% (n = 24), 16% (n = 8), and 2% (n = 1) presented, respectively, poor, good, and very good school levels; and 34% (n = 17) of the participants displayed a familial history of major psychiatric disorders.

In the whole sample, the mean of the paranoia subscale was 34.6 (SD 17.5), whereas depression was 17.64 (SD 7.64) and anxiety was 49.88 (SD 19.77). Measures of depression and anxiety had a moderate correlation between them (r = 0.598),

### 3.2. Content of Paranoid Thoughts

Table 1 shows the contents of paranoid thoughts analyzed with DoT. We found that 58% (n = 29) of the sample knew the person who was trying to harm them. Of the sample, 52% (n = 26) evaluated the threat as extremely powerful (median 7). The majority of the participants (34%, n = 17) believed the threat was imminent and 62% (n = 31) and 32% (n = 16) placed the threat outside the home or both outside/inside the home. The mean hours of being under threat were 5.60 (SD 5.28), and 8% of the participants (n = 4) responded that they spent between 12 to 24 h under threat. The mean of sureness regarding the possibility of the threat was 61.80 (SD 33.02), and 12% (n = 6) endorsed the maximum score (100). Of the adolescents, 62% (n = 31) and 66% (n = 33) thought that the threat was extremely awful and unfair, and the medians were, respectively, 7, and 7.50. On the other hand, 62% (n = 31) of the participants believed that they would cope well with the threat (median 7), and 60% (n = 30) believed that the threat was not deserved (median 2). Only 16% (n = 8) thought that the threat was certainly deserved. Rescue factors (beyond and under control) were highlighted, respectively, by 48% (n = 24) and 38% (n = 19) and the medians were 6 and 5.

Table 2 shows the types of harm beliefs. Of the sample, 30% (n = 15) reported someone attempting to beat them (physical harm), 12% (n = 6) that someone attempted to insult them (verbal harm), 8% (n = 4) that someone attempted to make them eat to get fat. We found that 4% (n = 2) endorsed psychological harm. Other harm beliefs were attempts to blame, became fat, separated from friends, control, and follow. We found 30% (n = 15) responded “do not know”.

All DoT items were positively and significantly correlated with dimensional paranoia except for coping, unfairness, help seeking, and control (Table 1).

### 3.3. Relationship between Paranoid Thoughts and Bullying Victimization

The partial correlation analysis between the content of persecutory symptoms and bullying victimization displayed the following results: the certainty of knowing who is the threat correlated with the total (0.444), physical (0.394), and verbal (0.394) victimization, meaning that adolescents who were more confident of knowing who was threatening them had higher scores on these specific forms of victimization. The powerfulness of the threat correlated with the verbal victimization (0.295), explaining that the greater the verbal victimization, the greater the threat score. The imminence of the threat was linked to the total (0.334) and verbal (0.399) victimization; therefore, higher scores on these forms of victimization indicated higher scores on the perception of the imminence of the threat. The hours under threat correlated with the verbal (0.415) victimization (adolescents with higher scores on verbal victimization spent more hours under threat). The sureness of the threat had a moderate correlation with total (0.387), physical (0.359), and verbal (0.443) victimization; therefore, all these forms of bullying victimization increased the security of the threat. On the other hand, the awfulness of the threat was linked to the total (0.336) and social manipulation (0.325) victimization with these higher victimizations indicating greater unpleasantness. No other correlations were found between other content of paranoid thoughts (coping deservedness, unfairness, and rescue factors). All the results are presented in Table 3.

## 4. Discussion

We described in detail the contents of the paranoid symptoms related to the bullying–victimization phenomenon in a help-seeking cohort of adolescents, who were screened positive for PLEs. As expected, our results pointed out that paranoid thoughts were frequent and intense in the PLE positive population, with concurrent high scores in depression and anxiety symptoms; this revealed that affective and psychotic dimensions were concurrent and non-competing in the sample. Several authors described a connection between the old-fashioned terms of neurosis and psychosis, overcoming the traditional dichotomy between the two constructs, and our results fit this view [23,26,27]. The analysis of the contents of paranoid thoughts showed intriguing results. In our sample, more than half of the adolescents reported that they knew the person who was threatening them and that the threat was very powerful. One third of the individuals believed that the threat was imminent and that it would occur outside the home (thus, judging the home as a safer place). On average, adolescents felt themselves under threat for approximately 5 h per day, and four individuals had very high threat intensity (12 to 24 h). The sureness of the threat was high in the sample, confirming the delusional nature of the thoughts. 

More than 60% of the participants judged the threat as awful and unfair, but also that they would cope with the threat and that it was not deserved. Half of the sample answered positively regarding the questions on rescue factors. As pointed out by Freeman, it is important to have a single symptom approach with a focus of the study of delusional beliefs. In fact, assessing more of the contents of delusional symptoms in detail may help in changing belief convictions with greater success and efficacy during the psychotherapeutic process [22]. From a psychopathological point of view, studying the content of paranoid symptoms in more detail makes paranoia itself not just a label, but a container full of information that can also be linked to the adolescent’s life experience. In this regard, our results on the link with bullying victimization displayed interesting implications. 

Physical victimization (i.e., kicks and punches; repeated and intense) increased the awareness of the person who was threatening and the security of the threat. This is not surprising as it is a type of direct victimization. Verbal victimization was linked to the certainty of knowing the person who was threatening, to the power, imminence, intensity, and security of the threat. Verbal bullying was typically the most common form of victimization [14], mostly associated with low self-esteem and self-efficacy, interpersonal concern, and social anxiety, which can be precursors of paranoia and, in particular, of these specific contents of paranoid symptoms [27]. Social manipulation, an indirect form of victimization (i.e., the exclusion of peers and spreading rumors), increased the awfulness of the threat: intuitively, having suffered from this type of situation can make the contents of the thinking about the threat very unpleasant.

This study has some theoretical and clinical implications. As already suggested, psychotic symptoms, and, specifically, paranoid thoughts, were experienced with an extreme variance of contents [13]. A detailed description of the delusional thinking helps with the understanding of the inner meaning, that, in turn, may be very helpful during the psychotherapeutic process. In fact, the possibility of better understanding the content of emerging or consolidated delusional thoughts and eventually linking them with past experiences [12] may make these experiences less frightening, more acceptable, and, finally, less distressing. In terms of early identification, it may also be useful to know that previous adverse experiences not only increased the risk but also shaped the content and nuanced certain details (the imminence, awfulness, power, and intensity of threat) of the paranoid thoughts. 

Our findings, although from a different perspective, are also in line with several others that posit the need of preventing bullying victimization in order to reduce mental health risks and psychotic disorder burdens among adolescents [18]. In the educational context, teachers and families must be very careful in observing and facing any bullying victimization situations and, in particular, they should evaluate the type of victimization and any associated psychological symptoms. Therefore, they must possess the skills to identify signs of their child’s bullying victimization. On the other hand, parental protective measures in peer violence prevention could include: everyday talks, family common meals, spending leisure time together, etc. Adolescents can benefit from educational interventions aimed at interpreting the threat and its characteristics in the light of the previous episodes of bullying. School psychologists could promote primary and secondary prevention with information, campaigns and therapeutic interventions designed to make the experiences less unpleasant and more understandable.

This study should be viewed in the light of several limitations and, thus, considered as a pilot. The sample size was relatively small and this limits the generalizability of the study; thereby, the conclusions must be carefully formulated. To improve this limit, we selected standardized measures. The cross-sectional design limits the understanding of the causal direction between bullying and paranoid thoughts (and their details); also, data were collected through self-report questionnaires, which expose the findings to social desirability bias. Furthermore, people with PLEs may suffer from recall bias.

In conclusion, many studies have highlighted the importance of quantitative data for psychosis and quantitative analyses for the relationship between psychotic symptoms and stressful events, such as bullying victimization. This study proposed a mixed method with a focus on qualitative data, alongside quantitative data. This enriches the information available with the specific contents of the paranoid thoughts and their relationship to victimization in a thematic continuity; this will improve our understandability of delusional phenomena. In summary, the powerfulness, imminence, sureness, and awfulness of the threat were correlated with the physical, verbal, and social manipulation victimization, and these relationships should be considered during the psychotherapeutic process of positive psychotic symptoms. The attempt to link the PLEs with a specific victimization experience can make the symptoms more understandable and increase coping strategies. From a social psychiatry perspective, the prevention of bullying is necessary. Interventions will be most effective if they target multiple environments, such as the family, school peers and teachers, and the community context. The characteristics of the bully, the victim, the bystanders, and the class context must be taken into consideration, as well as research-based programs on bullying prevention [28].

## Figures and Tables

**Table 1 behavsci-10-00122-t001:** Contents of paranoid thoughts: the Detail of Threat (DoT) descriptions and correlations with dimensional paranoia.

	N (%)	Mean (SD)	Median	Correlation with Dimensional Paranoia
DoT_1: Do you know who it is that is trying to harm you?				
No	13 (26%)			
Maybe	8 (16%)			
Yes	29 (58%)			0.584 *
DoT_2: How powerful is the person(s) trying to harm you?				
No	12 (24%)			
Maybe	12 (27%)			
Yes	26 (52%)		7	0.545 *
DoT_4: When do you think the harm is most likely to happen?				
6 months or longer	4 (8%)			
1 month to 6 months	4 (8%)			
1 week to1 month	5 (10%)			
0 to 7 days	20 (40%)			
It has been happening recently	17 (34%)			0.376 *
Dot_5: Where will the harm most likely occur?				
Inside my home	3 (6%)			
Outside my home	31 (62%)			
Both in and outside of my home	16 (32%)			0.079
DoT_6: In the 24 h of a day, how many of these hours are you under threat?		5.60 (5.28)		0.596 *
DoT_7: How sure are you that the harm is happening? Please give a percentage estimate of the strength of your belief (0–100%)		61.80 (33.02)		0.483 *
DoT_8: If the threat did happen, how awful would it be?				
No	4 (8%)			
Maybe	15 (30%)			
Yes	31(62%)		7	0.293 **
DoT_9: How well would you cope if the threat did occur?				
No	7 (14%)			
Maybe	14 (28%)			
Yes	29 (58%)		7	−0.023
DoT_10: Sometimes, people who think harm is going to happen think that they may deserve this harm. Do you feel as if you deserve to be harmed in the way you have talked about?				
No	30 (60%)			
Maybe	12 (24%)			
Yes	8 (16%)		2	0.267
DoT_11: How unfair is it that this is occurring to you?				
No	5 (10%)			
Maybe	12 (24%)			
Yes	33 (66%)		7,50	0.078
DoT_12: How likely is it that factors beyond your control could lead to you being rescued from this harm? For example, something to do with the person trying to harm you or something to do with other people that may result in the threat not occurring.				
No	9 (18%)			
Maybe	17 (34%)			
Yes	24 (48%)		6	−0.038
DoT_13: Overall, how much control do you have over the situation?				
No	7 (14%)			
Maybe	24 (48%)			
Yes	19 (38%)		5	−0.141

* <0.01, ** <0.05.

**Table 2 behavsci-10-00122-t002:** Content of harm beliefs.

50 Participants	Types of Ham Beliefs
1	Someone attempted to make me feel angry
2	Someone attempted to separate me from friends
15	Someone attempted to beat me
1	Someone attempted to yell
1	Someone attempted to blame me (for something bad)
4	Someone attempted to make me eat to become fat
6	Someone attempted to insult me
1	Someone attempted to control me
2	Someone attempted to cause me psychological harm
1	Someone attempted to steal my girlfriend/boyfriend
1	Someone attempted to follow me
15	Do not know

**Table 3 behavsci-10-00122-t003:** Partial correlation between the contents of paranoid thoughts and bullying victimization.

Controlled for Depression and Anxiety	MPVS_TOT	MPVS_P	MPVS_V	MPVS_SM	MPVS_AP
DoT_1: Do you know who it is that is trying to harm you?					
No					
Maybe					
Yes	0.444 *	0.394 *	0.434 *	0.216	0.222
DoT_2: How powerful is the person(s) trying to harm you?					
(0–10)	0.230	0.241	0.295 **	0.146	−0.057
DoT_4: When do you think the harm is most likely to happen?					
6 months or longer					
1 month to 6 months					
1 week to 1 month					
0 to 7 days					
It has been happening recently	0.334 **	0.216	0.399 *	0.117	0.257
Dot_5: Where will the harm most likely occur?					
Inside my home					
Outside my home					
Both in and outside of my home					
(0–10)	−0.130	−0.104	−0.125	−0.128	−0.008
DoT_6: In the 24 h of a day, how many of these hours are you under threat?					
(0–24)	0.226	0.047	0.415 *	0.106	0.087
DoT_7: How sure are you that the harm is happening? Please give a percentage estimate of the strength of your belief					
(0–100%)	0.387 *	0.359 **	0.443 *	0.115	0.185
DoT_8: If the threat did happen, how awful would it be?					
(0–10)	0.336 **	0.210	0.202	0.325 **	0.253
DoT_9: How well would you cope if the threat did occur?					
(0–10)	−0.129	−0.009	−0.158	−0.071	−0.102
DoT_10: Sometimes, people who think harm is going to happen think that they may deserve this harm. Do you feel as if you deserve to be harmed in the way you have talked about					
(0–10)	0.030	0.098	0.017	0.037	−0.014
DoT_11: How unfair is it that this is occurring to you?					
(0–10)	0.090	−0.49	0.191	−0.104	0.135
DoT_12: How likely is it that factors beyond your control could lead to you being rescued from this harm? For example, something to do with the person trying to harm you or something to do with other people that may result in the threat not occurring.					
(0–10)	−0.042	0.017	−0.079	0.042	−0.036
DoT_13: Overall, how much control do you have over the situation?					
(0–10)	−0.209	−0.045	−0.164	−0.078	−0.278

* <0.01, ** <0.05, MPVS_TOT: multidimensional peer victimization scale_total score, MPVS_P: multidimensional peer victimization scale_physical score, MPVS_V: multidimensional peer victimization scale_verbal score, MPVS_SM: multidimensional peer victimization scale_social manipulation score, MPVS_AP: multidimensional attack on property.

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
