# Peer review of "Details of the Contents of Paranoid Thoughts in Help-Seeking Adolescents with Psychotic-Like Experiences and Continuity with Bullying and Victimization: A Pilot Study"

_behavsci, 2020, doi:10.3390/bs10080122_

Round 1
Reviewer 1 Report
I am grateful that you have sent me the manuscript for review, I have really enjoyed reading it. Despite being limited to a descriptive analysis, it is an interesting topic, especially the perspective of continuity between the variables that are addressed.
I have some recommendations for authors that I hope can help improve the manuscript:
Provide data on the reliability of the scales after their application in the study sample.
Presentation of results ... it would be more appropriate to integrate the% in the text and indicate in parentheses the frequency of corresponding cases (n = ...). On the other hand, when presenting the value of a statistic in the text, it is recommended to put it before the value (r = ..., p <0.001), for example.
In Table 1 ... use lowercase for the frequency, when referring to a subsample, and not the total sample.
In the conclusions ... the authors should emphasize developing the practical implications, especially taking into account the educational context that is where bullying episodes usually occur. It is necessary to be clear about the possibilities offered by these results, in order to design adequate intervention programs, and also to indicate who the benefactors of these are (adolescents, families, teachers, ...).
Author Response
REVIEWER 1
I am grateful that you have sent me the manuscript for review, I have really enjoyed reading it. Despite being limited to a descriptive analysis, it is an interesting topic, especially the perspective of continuity between the variables that are addressed.
Reply: Thank you for the kind comment.
I have some recommendations for authors that I hope can help improve the manuscript:
Provide data on the reliability of the scales after their application in the study sample.
Reply: We are grateful to the reviewer for highlight this point. We modified the paragraph adding the following sentences:
1)The APSS had good predictive power, sensitivity and specificity, the score of 2 or more had sensitivity of 70% and specificity of 82.6%. The item with the better predictive power was the question ‘‘Have you ever heard voices or sounds that no one else can hear?’’ with a positive predictive power of 71.4%, and an non positive predictive power of 90.4% (screening instrument, page 2, line 70);
2) … showed very good psychometric properties: Cronbach’s α of 0.93 [15]and Cronbach’s α of 0.85 [16](dimensional paranoia, page 2, line 85);
3) (Cronbach’s α = .80) (Depression, page 2, line 90);
4) (internal variability for the MASC was in the excellent range = .6 - .9) (anxiety, page 3, line 112);
5) (Internal consistency: Physical victimization = 0.85; Verbal victimization = 0.75; Social manipulation = 0.77; Property attacks = 0.73) (bullying victimization, page 3, line 119).
Presentation of results ... it would be more appropriate to integrate the% in the text and indicate in parentheses the frequency of corresponding cases (n = ...). On the other hand, when presenting the value of a statistic in the text, it is recommended to put it before the value (r = ..., p <0.001), for example.
Reply: We agree with the reviewer. We modified as suggested in the result section.
In Table 1 ... use lowercase for the frequency, when referring to a subsample, and not the total sample.
Reply: Sorry for this. We did not use sub-sample. We made a mistake in showing the results, the dot 4 frequency (of the response 0 to 7 days) is 20 (40%) and not 10 (20%). We corrected the error in the table.
In the conclusions ... the authors should emphasize developing the practical implications, especially taking into account the educational context that is where bullying episodes usually occur. It is necessary to be clear about the possibilities offered by these results, in order to design adequate intervention programs, and also to indicate who the benefactors of these are (adolescents, families, teachers, ...).
Reply: Thank you the reviewer for this comment. We added these sentences to the discussion section (line 356): In educational context teachers and families must be very careful in observing and facing any bullying situations and in particular they should evaluate the type of victimization and any associated psychological symptoms. Adolescents can benefit from educational interventions aimed at interpreting the threat and its characteristics in the light of the previous episodes of bullying. School Psychologists could promote primary and secondary prevention with information campaign and therapeutic intervention in order to make the experiences less unpleasant and more understandable.
Reviewer 2 Report
Language
Proofreading should be done to improve and correct errors in English. Some of these are listed below:
Line 12: the certainty of threat correlate with physical
Line 17: Conclusion
Line 13: population people
Line 19-20: Please rewrite the sentence: In a social psychiatry perspective the prevention of bullying is mandatory.
Line 40: In this framework that mixes a social perspective into a developmental psychopathology context
Line 46: in an adolescent cohort of help seeking adolescent with PLEs
Line 46: The secondary aim
Line 59: methodologies of the research
Line 62-63: Screening instrument: The Adolescent Psychotic Like Symptom Screener (APSS) constituted 62 the screening instrument for this study
Line 63: seven items questionnaire
Line 64: >2 point
Line 102-103: socio-demographics characteristics
A brief summary
The aim of the paper was to describe the content of paranoid symptoms and to exam the relationships with experience of bullying victimization in help seeking adolescents with PLEs.
The study proposed a mixed method of qualitative and quantitative data analysis. Gives the description about the specific contents of the paranoid thoughts and their relationship with experience of bullying victimization to understand the delusional phenomena.
The conclusions based on the findings of this study contain worthwhile suggestions for adolescents’ psychotherapists
Broad comments
Since the study is about the content of paranoid thoughts and relationships with bullying victimization experiences I would suggest to change slightly the title of the manuscript to make it more specific. Considering the study limitations (line 218), the information about pilot study should be added.
In Abstract, the variables and study tools are missed, and should be written here.
The study shows results of the analysis of the data gathered in small size study group of 50 adolescents, so the conclusions should be carefully formulated.
The harmful impacts of bullying victimization in adolescents’ mental health has been broadly studied and this is evident (Hawker, D. S. J., & Boulton, M. J. (2000, May). Twenty years’ research on peer victimization and psychosocial maladjustment: A meta-analytic review of cross-sectional studies. Journal of Child Psychology and Psychiatry and Allied Disciplines. https://doi.org/10.1017/S00219630990055450).
So in Introduction and in Conclusions it would worthwhile to write more in advance than current knowledge about adolescents mental health prevention and promotion in relation to bullying experience.
The suggestion’s for school psychologists for primary and secondary bullying prevention in adolescents on individual and school level would be useful too.
Specific comments
Abstract
Line 5 and 25: for other than social sciences audience would be better to specify the term of “environmental” as “social environmental”
Line 9-10 In Method: Number, age, sex of participants should be in this section. Please write also about analyzed variables/factors and the tools applied for data collection
Line10-17: In Results: No need to put p values
Line 19: Conclusions should be formulated to the wider readership of the Journal, including professionals working with adolescents, for example school psychologists
Keywords
Regarding the content of this study to make it more specific I would suggest to use: “adolescents” instead of ‘youths”, and “Psychotic Like Experiences” instead of “stress”, and “bullying victimization” instead of “bullying”
Introduction:
Regarding selected factors and variables for the analysis, the referred publications are poorly discussed
Line: 33, 34: Please specify as “bullying victimization” or “been bullied” or “victimization”
Line 37-38: Please specify, explain “early life adversities” and “specific psychotic symptoms”
Line 39: not clear why this subtitle is here
Line 41-42: The sentence is not clear. Please rewrite
2.1. Study design and sample
Please write what was the mean age of the sample
Please explain the study design written as “The setting was removed for double blind peer review”
2.2. Measures
Line 70: please write “questions” not “variables” here.
Results
For ordinal variables included in DoT it is not possible to count Mean values, but Median values.
The analysis should be corrected.
Line 13-136: reg. Table 2 it is not clear why there are comments about one belief with 2 answers , and no comments of 4 answers “to make me eat” and “do not know”.
Tables should be entitled more specific in detailed way.
Discussion
Please do not write about individuals, persons who participated in the study as “subjects”.
Please separate the part of the text about the study limitations.
Conclusions
Line 228-231 Please rewrite the sentence, and specify what kind of suggestions for psychotherapists and others for example school psychologists for primary prevention have resulted from the findings of the study.
References
Refernces do not meet the editor’s requirements reg. the style, and should be improved and corrected.
Author Response
REVIEWER 2
Language
Proofreading should be done to improve and correct errors in English. Some of these are listed below:
Line 12: the certainty of threat correlate with physical
Line 17: Conclusion
Line 13: population people
Line 19-20: Please rewrite the sentence: In a social psychiatry perspective the prevention of bullying is mandatory.
Line 40: In this framework that mixes a social perspective into a developmental psychopathology context
Line 46: in an adolescent cohort of help seeking adolescent with PLEs
Line 46: The secondary aim
Line 59: methodologies of the research
Line 62-63: Screening instrument: The Adolescent Psychotic Like Symptom Screener (APSS) constituted 62 the screening instrument for this study
Line 63: seven items questionnaire
Line 64: >2 point
Line 102-103: socio-demographics characteristics
Reply: Thank you the reviewer for highlights these language errors. We had a careful revision of the English language throughout the manuscript and we addressed specifically these points.
A brief summary
The aim of the paper was to describe the content of paranoid symptoms and to exam the relationships with experience of bullying victimization in help seeking adolescents with PLEs.
The study proposed a mixed method of qualitative and quantitative data analysis. Gives the description about the specific contents of the paranoid thoughts and their relationship with experience of bullying victimization to understand the delusional phenomena.
The conclusions based on the findings of this study contain worthwhile suggestions for adolescents’ psychotherapists
Broad comments
Since the study is about the content of paranoid thoughts and relationships with bullying victimization experiences I would suggest to change slightly the title of the manuscript to make it more specific. Considering the study limitations (line 218), the information about pilot study should be added.
Reply: We agree with the reviewer and we modified the title as follow: Details of the content of paranoid thoughts in help-seeking adolescents with Psychotic Like Experiences and continuity with bullying victimization: a pilot study.
In Abstract, the variables and study tools are missed, and should be written here.
Reply: We agree with the reviewer and we added this sentence in the abstract: (paranoia, the specific psychotic experience questionnaire SPEQ; content of paranoid thoughts the detail of threat DoT; bullying victimization, the Multidimensional peer victimization scale MPVS, depression, the Children Depression Inventory CDI; Anxiety, the Multidimensional Anxiety Scale MASC).
The study shows results of the analysis of the data gathered in small size study group of 50 adolescents, so the conclusions should be carefully formulated.
Reply: This is an important point. We added this sentence at the end of the manuscript (line 365): and this limited the generalizability of the study so that the conclusions must be carefully formulated, to improve this limit we selected standardized measures
The harmful impacts of bullying victimization in adolescents’ mental health has been broadly studied and this is evident (Hawker, D. S. J., & Boulton, M. J. (2000, May). Twenty years’ research on peer victimization and psychosocial maladjustment: A meta-analytic review of cross-sectional studies. Journal of Child Psychology and Psychiatry and Allied Disciplines. https://doi.org/10.1017/S00219630990055450).
Reply: We are very grateful to the reviewer for this suggestion. We included this quote in the manuscript (line 34)
So in Introduction and in Conclusions it would worthwhile to write more in advance than current knowledge about adolescents mental health prevention and promotion in relation to bullying experience.
The suggestion’s for school psychologists for primary and secondary bullying prevention in adolescents on individual and school level would be useful too.
Reply: We believe that this point could improve the manuscript. We added a sentence and a quote in the introduction (line 38): Prevention of bullying in the school is a sensible topic for promotion of mental health in adolescents. Anti-bullying interventions effectively reduce school-bullying (11); and in the discussion section (line 356): In educational context teachers and families must be very careful in observing and facing any bullying situations and in particular they should evaluate the type of victimization and any associated psychological symptoms. Adolescents can benefit from educational interventions aimed at interpreting the threat and its characteristics in the light of the previous episodes of bullying. School Psychologists could promote primary and secondary prevention with information campaign and therapeutic intervention in order to make the experiences less unpleasant and more understandable.
Specific comments
Abstract
Line 5 and 25: for other than social sciences audience would be better to specify the term of “environmental” as “social environmental”
Line 9-10 In Method: Number, age, sex of participants should be in this section. Please write also about analyzed variables/factors and the tools applied for data collection
Line10-17: In Results: No need to put p values
Line 19: Conclusions should be formulated to the wider readership of the Journal, including professionals working with adolescents, for example school psychologists
Reply: We changed the manuscript in light of these comments. For the last comment see above
Keywords
Regarding the content of this study to make it more specific I would suggest to use: “adolescents” instead of ‘youths”, and “Psychotic Like Experiences” instead of “stress”, and “bullying victimization” instead of “bullying”
Reply: we changed the keywords as suggested
Introduction:
Regarding selected factors and variables for the analysis, the referred publications are poorly discussed
Line: 33, 34: Please specify as “bullying victimization” or “been bullied” or “victimization”
Reply: we add the sentence: that consists of intentional and repeated aggressive acts in a context of imbalance of power (line 33)
Line 37-38: Please specify, explain “early life adversities” and “specific psychotic symptoms”
Reply: we add the sentence: such as bullying victimization(line 37) and such as paranoia and hallucination (line 37-38)
Line 39: not clear why this subtitle is here
Reply: sorry for this. This is not a subtitle (Paragraph removed for double blind peer review). It is a sentence that explains that in this point there is a brief reference to our previous study which dealt with a similar topic. Since the results of the study and the bibliographic source are present, we have removed it for peer review
Line 41-42: The sentence is not clear. Please rewrite
We changed the sentence as follow: In this framework that mixes a social perspective and developmental psychopathology,
2.1. Study design and sample
Please write what was the mean age of the sample
Reply: we inserted the mean age of the sample in the results and method section
Please explain the study design written as “The setting was removed for double blind peer review”
Reply: the setting was the place of the study and we removed it for the double blind peer review process
2.2. Measures
Line 70: please write “questions” not “variables” here.
Reply: we changed as suggested
Results
For ordinal variables included in DoT it is not possible to count Mean values, but Median values.
Reply: we changed the mean values with median values
The analysis should be corrected.
Line 13-136: reg. Table 2 it is not clear why there are comments about one belief with 2 answers , and no comments of 4 answers “to make me eat” and “do not know”.
Tables should be entitled more specific in detailed way.
Reply: we adjusted this section and commented also other replies
Discussion
Please do not write about individuals, persons who participated in the study as “subjects”.
Reply: we changed the term suggested
Please separate the part of the text about the study limitations.
Reply: We separated the study limitation from the text
Conclusions
Line 228-231 Please rewrite the sentence, and specify what kind of suggestions for psychotherapists and others for example school psychologists for primary prevention have resulted from the findings of the study.
Reply: we rewrite the sentence as follow: In summary, powerful, imminence, sureness and awfulness of the threat correlated with physical, verbal and social manipulation victimization and these relationships should be taken into account during psychotherapeutic process of positive psychotic symptoms. In fact the attempt to link the PLEs with a specific victimization experience can make the symptoms more understandable and increase coping strategies. In a social psychiatry perspective, the prevention of bullying is necessary.
References
References do not meet the editor’s requirements reg. the style, and should be improved and corrected.
reply: Thank you, we adjusted references
Round 2
Reviewer 2 Report
Language
Few grammar errors in writing should be corrected, especially the names and titles with capital or small letters.
A brief summary
The aim of the paper was to describe the content of paranoid symptoms and to exam the relationships with bullying victimization in help seeking adolescents with PLEs.
The study proposed a mixed method of qualitative and quantitative data analysis. Gives the description of the specific contents of the paranoid thoughts and their relationship with experience of bullying victimization to understand the delusional phenomena. The conclusions contain worthwhile suggestions for bullying prevention measures regarding adolescents’ mental health impacts.
Broad comments
The revised improved version of the manuscript has got much higher quality, value and impact. The revision is more accurate, clear and better contextualized.
The Abstract includes the variables and study tools, and is well elaborated.
The conclusions contain suggestion’s for bullying prevention programs including parents and teachers, and these measures may prevent and reduce psychotic disorders among adolescents.
Considering the all text, I would appreciate not to write about members of the study group, individuals, persons who participated in the study as “subjects” (line: 138, 145, 150,163).
In Abstract it would be good formulated conclusions to the wider readership of the Journal, including professionals working with adolescents, for example school psychologists, teachers, who communicate also with parents.
The Introduction, regarding selected factors and variables for the analysis, this part seems to be quite modest.
Regarding mean age value it is very seldom to present this in number of months. Usually age is given in years.
Just few specific comments for Conclusions part:
Line: 234 Instead of „fighting” I would suggest to use „preventing”
Line 235: It would be good to specify the burden of the disease, which is “psychotic disorders burden among adolescents”. Teachers and parents should not only observe but also be skilled, capable to identify signs of their child’s bullying victimization. One of the most effective parental protective measures in peers’ violence prevention are: everyday talks, family common meals, leisure time spending, etc.
Line 262: The informative value of this sentence is poor. It would be good to develop it and to underlie the role of bullying prevention in mental disorders prevention among adolescents.
Author Response
Language
Few grammar errors in writing should be corrected, especially the names and titles with capital or small letters.
We corrected in the abstract and material and methods,
A brief summary
The aim of the paper was to describe the content of paranoid symptoms and to exam the relationships with bullying victimization in help seeking adolescents with PLEs.
The study proposed a mixed method of qualitative and quantitative data analysis. Gives the description of the specific contents of the paranoid thoughts and their relationship with experience of bullying victimization to understand the delusional phenomena. The conclusions contain worthwhile suggestions for bullying prevention measures regarding adolescents’ mental health impacts.
Broad comments
The revised improved version of the manuscript has got much higher quality, value and impact. The revision is more accurate, clear and better contextualized.
The Abstract includes the variables and study tools, and is well elaborated.
The conclusions contain suggestion’s for bullying prevention programs including parents and teachers, and these measures may prevent and reduce psychotic disorders among adolescents.
Considering the all text, I would appreciate not to write about members of the study group, individuals, persons who participated in the study as “subjects” (line: 138, 145, 150,163).
We agree. We chenged in light of the reviewer’s comment. We used terms such as: participants, adolescents and sample
In Abstract it would be good formulated conclusions to the wider readership of the Journal, including professionals working with adolescents, for example school psychologists, teachers, who communicate also with parents.
We added the following sentence: Teachers and family must actively monitor early signs of bullying victimization and school psychologist could promote preventive and therapeutic intervention
The Introduction, regarding selected factors and variables for the analysis, this part seems to be quite modest.
In the introduction section we removed a brief paragraph in which we showed information that could alter the double blind peer review process. We reformulated the paragraph and in continuity with this paragraph we added the following: To continue on this path we used a model that includes not only the presence of paranoid thoughts but also the content of those thoughts. Freeman et al. in order to assess thedetails of the delusional thoughts (e.g. conviction, distress, the power and the imminence of threat) designed a new questionnaire “the Detail of Threat”; in their study authors found that several features of the content of the delusional thoughts were associated with depression, self-esteem and anxiety. This questionnaire is complementary to a more quantitative one on the presence of paranoid symptoms such as the paranoia scale of the specific psychotic experiences questionnaire (SPEQ) [15]. There are several questionnaires that assess bullying victimization [16]. The multidimensional peer victimization scale [17]is a simple, fast and reliable analysis tool already used in several studies [18,19].
Regarding mean age value it is very seldom to present this in number of months. Usually age is given in years.
We have added the age in years in brackets (14 years and 1 month)
Just few specific comments for Conclusions part:
Line: 234 Instead of „fighting” I would suggest to use „preventing”
We changed the term suggested.
Line 235: It would be good to specify the burden of the disease, which is “psychotic disorders burden among adolescents”. Teachers and parents should not only observe but also be skilled, capable to identify signs of their child’s bullying victimization. One of the most effective parental protective measures in peers’ violence prevention are: everyday talks, family common meals, leisure time spending, etc.
Thank you for this comment, we changed the paragraph as follow: Also, our findings, although from a different perspective, are in line with several others that posit the need of preventing bullying victimization in order to reduce mental health risk and psychotic disorders burden among adolescents [18]. In educational context teachers and families must be very careful in observing and facing any bullying victimization situations and in particular they should evaluate the type of victimization and any associated psychological symptoms. Therefore they must possess the skills to identify signs of their child’s bullying victimization. On the other hand, parental protective measures in peers’ violence prevention could be: everyday talks, family common meals, leisure time spending, etc.
Line 262: The informative value of this sentence is poor. It would be good to develop it and to underlie the role of bullying prevention in mental disorders prevention among adolescents.
We agree and we added the following sentence: Interventions will be most effective if they targetmultiple environments such as family, school peer and theachers and community context. The characteristics of the bully, the victim, the bystanders and the class context must be taken into consideration as well as research-based programs on bullying prevention [28].